# Continuous Hue-Based Self-Calibration of a Smartphone Spectrometer Applied to Optical Fiber Fabry-Perot Sensor Interrogation

**DOI:** 10.3390/s20216304

**Published:** 2020-11-05

**Authors:** Aleksandr Markvart, Leonid Liokumovich, Iurii Medvedev, Nikolai Ushakov

**Affiliations:** 1Institute of Physics, Nanotechnology and Telecommunications, Peter the Great St. Petersburg Polytechnic University, 195251 St. Petersburg, Russia; leonid@spbstu.ru; 2Institute of Systems and Robotics, University of Coimbra, Rua Silvio Lima—Polo II, 3030-290 Coimbra, Portugal; iurii.medvedev@isr.uc.pt

**Keywords:** optical fiber sensor, smartphone-based sensor interrogation, image processing, Fabry-Perot interferometer, spectral calibration, spectral interferometry, white-light interferometry, smartphone optical spectrometer.

## Abstract

Smartphone-based optical spectrometers allow the development of a new generation of portable and cost-effective optical sensing solutions that can be easily integrated into sensor networks. However, most commonly the spectral calibration relies on the external reference light sources which have known narrow spectral lines. Such calibration must be repeated each time the fiber and diffraction grating holders are removed from the smartphone and reattached. Moreover, the spectrometer wavelength scale can drift during the measurement because of the smartphone temperature fluctuations. The present work reports on a novel spectral self-calibration approach, based on the correspondence between the light wavelength and the hue features of the spectrum measured using a color RGB camera. These features are caused by the nonuniformity of camera RGB filters’ responses and their finite overlap, which is a typical situation for RGB cameras. Thus, the wavelength scale should be externally calibrated only once for each smartphone spectrometer and can further be continuously verified and corrected using the proposed self-calibration approach. An ability of the plug-and play operation and the temperature drift elimination of the smartphone spectrometer was experimentally demonstrated. Conducted experiments involved interrogation of optical fiber Fabry-Perot interferometric sensor and demonstrated a nanometer-level optical path difference resolution.

## 1. Introduction

Optical fiber sensors (OFS) are an emerging research topic, gaining a lot of attention from both academia and industry. During the last three decades they have progressed from simple laboratory prototypes to high-end commercial products, forming a several billion dollar market. OFS are an ideal choice for various sensing tasks, including biomedical, oil and gas, avionics, structure health monitoring and others. Their advantages are small size, biocompatibility, chemical neutrality, ability to perform remote measurements and multiplex several sensors, electromagnetic neutrality and absence of electric currents or radiofrequency fields in the sensing element. Another advantage is the ability to either make the sensing element out of a section of a fiber, or to form a sensing element with free-space propagation of light, whilst fiber acts as a light feeding element.

One of the most commonly used interrogation approaches for OFS is spectral interrogation, which includes measurement of a spectral function of an OFS (either reflective or transmissive) and its further processing. This approach is used to acquire signals from OFS based on surface plasmon resonance (SPR) [1], grating-based sensors [2,3] and interferometric sensors [4,5]. Spectral interrogation of interferometric sensors is typically referred to as spectral interferometry or white-light interferometry.

Spectral interferometry is widely used for interrogation of interferometric OFS with small (typically less than 1 mm) optical path differences (OPD), typically extrinsic Fabry-Perot interferometers (EFPI) [6] or in-fiber intermode interference sensors [7]. This approach offers high resolutions [6,8], ability to multiplex several sensors [9,10] and to perform absolute measurements. However, the accuracy of the spectral interferometric techniques is inextricably linked to the accuracy and stability of the spectrometer used to measure the interferometer spectrum [4]. Various spectral calibration methods have been proposed so far [11,12,13], most of them are based on complex modifications of the sensing scheme, including reference interferometers, polarization diversity and others.

Shortly after the smartphones became widespread and deeply integrated into our everyday routines, it was proposed to use them as the core elements of portable optical spectrometers [14,15], which can be applied to interrogation of optical sensors [16,17,18] and OFS in particular [19]. This relatively narrow research field is rapidly emerging, with examples of intensity-based [20], SPR-based [19,21,22], EFPI [23] and chirped FBG-based [24] sensors interrogation reported so far. These systems typically use smartphone flash LED as a light source and smartphone camera as a photodetector array. Even by using cheap plastic diffraction gratings, usually made of a CD or DVD disk piece, it is possible to make a smartphone-based optical spectrometer, which characteristics are comparable with the commercially available spectrometers [14,15,25,26,27,28,29,30].

Spectral calibration methods are of particular importance for smartphone-interrogated optical fiber sensors with spectral interrogation since the additional constructions holding fibers and diffraction gratings usually can not be robustly fixed on a smartphone and are typically fixed on it with some pegs or clamps [24,28]. Therefore, the position of spectrum projected on the smartphone camera matrix might change in case of mechanical distortions, if the system is hand-held and is operating on the go, or when the fiber and diffraction grating holders are removed during transportation and reattached. Change of ambient conditions, especially temperature can cause thermal expansion or shrinking of fiber and diffraction grating holders, also resulting in spectral shift. Nevertheless, despite the great importance of these aspects for measurement accuracy, to the best of our knowledge, neither of them has been considered in the literature dedicated to smartphone-based spectrometers and optical sensors. Therefore, continuous spectral calibration methods able to run alongside demodulation of the target signal from the measured spectra will significantly expand potential applications of portable optical fiber sensing systems with smartphone-based interrogation.

To date, we are aware of an attempt to use the hue (H) information of the spectrum captured with a color smartphone camera to evaluate the wavelength scale of the measured spectrum [31], where the authors have based their calibration approach on a linear fitting of H(λ) function. However, on the one hand, the linear slope of H(λ) lacking any steep features is not the best candidate for accurate wavelength calibration and on the other hand, as will be revealed in our work, the H(λ) is indeed nonlinear; thus, more complex models must be used for its description. Another example of using the hue of measured image in a smartphone spectrometer can be found in [32], where it was used to classify analytes. Yet, there is not much light shed in [32] onto the physical principles behind the proposed sensing principle, as a trained convolutional neural network was used to perform the classification according to the image hue distribution.

Actually, the use of color camera for optical spectrum measurement is quite intuitive and logical, since color camera already contains spectral resolving elements, which, however, are not suited for high-precision spectral discrimination, in contrast with diffraction gratings. Nevertheless, a way of exploiting these spectral elements and the spectrum features they cause for the purpose of spectral calibration can be found. In the current work, we have developed a spectral calibration approach similar to [31]; yet, in order to improve the calibration accuracy, we have carefully analyzed the measured optical spectra and revealed that there are characteristic features present in the measured spectra. We have developed a wavelength self-calibration method able to operate continuously and provide a reliable wavelength scale reference of the measured spectra. For that, we have identified two reference points in the hue distribution of the measured spectrum, whose positions in the optical spectrum can be measured and used further to deduce the wavelength scale of the measured spectrum.

## 2. Smartphone-Based Optical Fiber Sensor Interrogation System

The concept of the smartphone-based interrogation system, similar to most of the already known smartphone-based spectrometers (SPBS) [19,23,25], is shown in Figure 1a. The photo of the experimental setup is shown in Figure 1b. The smartphone LED was used as a light source. SPBS was assembled using the smartphone camera, a piece of a DVD disk acting as a diffraction grating and a 150 μm width slit consisting of two halves of a razor blade. These components were fixed on a 3D-printed holder, attached to the smartphone with a peg. A detailed description of the developed SPBS can be found in our previous work [24]. The only difference of the current work is the use of a color camera instead of the monochrome one in [24].

The smartphone used in the proposed OFS interrogation setup was Huawei P20 Pro, featuring a 10-megapixel color camera. Most of the experiments were conducted using this model. However, several other widespread smartphone models were also tested and turned out to be suitable for realization of the proposed spectral self-calibration approach as well.

Since spatially incoherent light irradiated by the smartphone LED cannot be efficiently coupled to singlemode fibers due to fundamental reasons [33,34], it is only possible to use multimode fibers with the smartphone-based systems. Therefore, graded index multimode fibers with a core diameter of 62.5 μm were used for guiding light to and from the interferometric sensing element, which was connected to the smartphone flash LED and camera via a multimode 50:50 coupler. The 60 degrees tilt of the output fiber through which the diffraction grating and the camera are irradiated was chosen so that only the first-order diffraction beam impinged the smartphone camera sensor.

The sensing element used in the performed experiments was an extrinsic fiber Fabry-Perot interferometer, formed by the end faces of two multimode fibers. The reflectivities of the fiber ends are about 3%, which is due to the Fresnel effect. Thus the interferometer was low-finesse and a two-beam approximation was used for signal processing. The fibers were terminated with FC-PC connectors, which were fixed inside a standard mating sleeve. The air gap $L_{0}$ between the fiber end faces, determining the interferometer OPD was set to about 8.8 μm. The OPD value was verified using spectral interferometric measurement with a commercial spectrometer Hamamatsu C10082CAH.

## 3. Proposed Spectral Calibration Approach

### 3.1. Analysis of Spectra Images Measured with Smartphone Spectrometer

An example of image of the smartphone LED spectrum measured with the smartphone camera is shown in Figure 2a. Exposure parameters were the following: ISO 50 (minimal value), shutter speed 1/17 s, aperture f/1.8, color temperature in camera settings was set to 3000 K. It can be seen even with a naked eye that the yellow and cyan areas of the spectrum (they actually are the transitions between the red, green and blue areas) are quite narrow, while the color hue distribution of blue, green and red areas is nearly uniform. On the other hand, uniformity of the wavelength scale of the assembled SPBS was verified by comparing the shapes of interferometer spectra measured with SPBS and a commercial spectrometer as described below. Therefore, this significant hue nonuniformity of the measured spectrum can be harnessed to provide a reliable spectrum self-calibration according to the two distinctive spectrum features.

These spectral features can be illustrated more directly by transforming the image shown in Figure 2a from RGB to hue, saturation, value (HSV) format and analyzing the hue channel. A cross-section of spectrum image’s hue distribution along the *v* axis at u=1400 is shown in Figure 2b; RGB representation of the same cross-section is shown in Figure 2c. By comparing the plots in Figure 2b,c, it can be clearly seen that the transition areas are due to the overlap between the spectral ranges occupied by red, green and blue channels and is determined by the optical filters used in the camera. The hue values are calculated according to the following equation
(1)$$H=\left\{\begin{array}{c} undefined,\;if\;MAX=MIN\\ \left(1/6\times \left(G-B\right)\cdot {\left(MAX-MIN\right)}^{-1}+1\right)\;mod\;1,\;if\;MAX=R\\ 1/6\times \left(B-R\right)\cdot {\left(MAX-MIN\right)}^{-1}+1/3,\;if\;MAX=G\\ 1/6\times \left(R-G\right)\cdot {\left(MAX-MIN\right)}^{-1}+2/3,\;if\;MAX=B \end{array}\right.,$$
where MAX and MIN denote the maximal and minimal values among the *R*, *G* and *B*. As follows from the Equation (Equation 1), the range of the hue values is [0, 1), where 0 corresponds to pure red, 1/3 to pure green and 2/3 to pure blue, while the intermediate values encode the combinations of these colors as shown in the color wheel in the inset in Figure 2b. This color representation allows to draw a direct analogy between the hue value and angle (or phase).

As can be seen from the color wheel in the inset in Figure 2b, the colors close to red can have hue values a bit greater than 0 as well as a bit smaller than 1, which can result in discontinuities of hue value distributions, as shown with a blue solid curve in Figure 2b. In order to eliminate these discontinuities, an operation similar to phase unwrapping [35] was applied to hue values—if a difference between adjacent hue values $H_{i}$, $H_{i+1}$ greater than 0.5 was detected, then the value of $H_{i+1}$ was corrected by replacing it with $H_{i+1}-1$ value. This unwrapping procedure was applied to all columns of the measured spectrum image. Unwrapped hue distribution is shown in Figure 2b with a dashed red curve.

It is verified by the plot in Figure 2b that the spectrum consists of three main areas with relatively uniform hue distributions, corresponding to red, green and blue and two transition areas, where hue value changes rapidly. The latter two areas will be used for spectral self-calibration by monitoring their positions by means of least-squares fitting.

However, as can be seen in Figure 2b, the change of hue value in these transition areas is nonlinear, which may result in fitting inaccuracies and calibration errors. Therefore, optimal color temperature setting of the camera must be found in order to simplify the calibration task and increase its accuracy. Optimal color temperature for the smartphone used in the experiments turned out to be 4000 K, the details on how it was evaluated can be found in Appendix A.

It should be noted that the spectrum image measured with a color camera is a result of a convolution and consequent multiplication of the true optical spectrum with two functions. The first operation is a convolution of the input optical spectrum with spectrometer instrument function (IF), which takes place for all spectrometers [36]. The second operation is multiplication of the convoluted input spectrum by the spectral responses of the optical filters in red, green and blue (RGB) channels of the camera. In the ideal case of a spectrometer with an infinitely narrow IF (corresponding to unrealistically high spectral resolution) the hue-to-wavelength correspondence of the measured spectrum will depend only on the RGB optical filters’ spectral responses and will be independent of an input signal optical spectrum.

However, for finite width IF (finite spectral resolution of spectrometer) it will not be the case—the hue value of each pixel will be affected not only by the input signal intensity at a particular wavelength, but rather by the signal intensity in some spectral neighborhood. The wider the IF, the more the Hue distribution will depend on the input signal spectrum. In the context of our work, this is an undesirable effect as it will cause smoothing of the hue distribution features, discussed above. In order to avoid this detrimental effect, we have assembled a smartphone spectrometer with a fairly high spectral resolution of 4 nm.

When the hue distribution is independent of the input signal spectrum and possesses clearly detectable features, these features can be unambiguously associated with the corresponding wavelength points. In other words, the wavelength scale of the measured spectrum will be linked to the spectral responses of the RGB optical filters, which are quite stable. A vital advantage of such correspondence is that for each smartphone it must be performed only once, after that the self-calibration can be performed for all subsequent measurements. The most crucial condition for robust operation of the proposed spectral self-calibration approach is that the analyzed optical spectrum must be sufficiently broad (covering both spectral features used for the wavelength scale calculation) and sufficiently bright for proper calculation of hue distribution of the spectrum image.

### 3.2. Spectra Alignment

Below, we describe the consecutive stages of the proposed spectral self-calibration approach, based on the spectra features described in the previous subsection.

(1)After acquisition of a spectrum image, it was transformed from RGB to HSV format. Then it was cropped in order to remove blank areas—columns which RMS values of the value channel were less than half of the maximal column’s RMS were removed, after that RMS position $\tau_{RMS}$ and width $w_{RMS}$ of rows’ RMS values distribution were calculated according to equations
(2)$$\tau_{RMS}=\frac{\sum x\cdot t}{\sum x},$$
(3)$$w_{RMS}=\sqrt{\frac{{\sum |x|}^{2}\cdot {\left(t-\tau_{RMS}\right)}^{2}}{{\sum |x|}^{2}}},$$
where *x* is the analyzed array, *t* is the indexing variable (row number in our case). The rows falling outside of the [$\tau_{RMS}-w_{RMS}$; $\tau_{RMS}+w_{RMS}$] interval were removed from the analysis.(2)At the next step, histogram of all hue channel values of the cropped spectrum image was calculated. The number of histogram bins $N_{b}$ was chosen according to the Freedman-Diaconis rule [37]
(4)$$N_{b}=\frac{\left(\max\left(H\right)-\min\left(H\right)\right)\cdot \sqrt{M}}{2\cdot IQR},$$
where *M* is the number of pixels in the cropped image, IQR is the interquartile range of *H* values. IQR is also known as midspread and is a difference between the first and the third quartiles of the considered data set. Typical number of bins estimated according to Equation (Equation 4) for the acquired and cropped spectra images was between 70 and 80.(3)After calculating the histogram, the coordinates of its bins $h_{r}$, $h_{g}$ and $h_{b}$ with greatest count numbers were found. These bins correspond to the majority of pixels in the green, red and blue spectrum areas, respectively and therefore were located in the neighborhoods of 0, 0.33 and 0.67 values. After that, RMS positions $h_{r_{RMS}}$, $h_{g_{RMS}}$ and $h_{b_{RMS}}$ and widths $\delta h_{r_{RMS}}$, $\delta h_{g_{RMS}}$ and $\delta h_{b_{RMS}}$ of histogram areas inside the intervals [$h_{r}-\delta h$, $h_{r}+\delta h$], [$h_{g}-\delta h$, $h_{g}+\delta h$], [$h_{b}-\delta h$, $h_{b}+\delta h$] were estimated according to Equations (Equation 2) and (Equation 3). Selecting δh=1/6, the whole range of possible hue values was accounted for.This allowed the red-to-green and green-to-blue transition areas to be identified in each column as the pixels, whose hue values lie within [$h_{r_{RMS}}-\delta h_{r_{RMS}}$, $h_{g_{RMS}}+\delta h_{g_{RMS}}$] and [$h_{g_{RMS}}-\delta h_{g_{RMS}}$, $h_{b_{RMS}}+\delta h_{b_{RMS}}$] intervals, respectively. After the transition areas were identified, their positions were estimated by fitting all of them by a tanh function of a form given below
(5)$$H=a\cdot \tanh\left(\left(t-t_{T}\right)/t_{\mathrm{dur}}\right),$$
where *a* and $t_{\mathrm{dur}}$ are amplitude and temporal scaling parameters, *t* is the indexing variable (row number in our case) and $t_{T}$ is the position of the middle of the transition area. The values of parameters *a*, $t_{\mathrm{dur}}$ and $t_{T}$ were found by least-squares fitting procedure. Since tanh is an odd function, all the fitted sections were subject to mean subtraction operation prior to the fitting.(4)The obtained dependencies of hue transition areas’ coordinates on *u* coordinate (spectrum column number) were approximated by second-order polynomial fits, allowing to eliminate some noise effects. The fits are shown in Figure 2a with red and blue dashed curves. Thanks to such an accurate determination of the hue transition areas’ coordinates, the circular distortion of measured spectra, clearly observable in Figure 2a, leading to relative shifts and change of wavelength sampling of spectrum columns could be compensated.(5)Finally, knowing the positions of two reference points, corresponding to certain wavelengths for each spectrum column, it was possible to interpolate all the columns and resample them to a true wavelength scale. The interpolation based on the fast Fourier transform [38] was applied. After spectrum columns interpolation, the columns were averaged, resulting in substantial increase of spectrum signal-to-noise ratio. Therefore, regardless of misalignment of the optical components of SPBS, the wavelength scale of the measured spectrum can be restored according to the two reference points.

Reproducibility of the proposed spectral self-calibration approach was tested using 7 smartphone models. The test results are presented in Appendix B.

### 3.3. Initial Wavelength-To-Hue Calibration

Calibration of the wavelengths corresponding to the red-to-green and green-to-blue transition areas was performed by measuring spectrum of a thermally and mechanically isolated Fabry-Perot interferometer with our developed SPBS and commercial spectrometer Hamamatsu C10082CAH. Smartphone LED was used as a light source for both measurements. Reference shape of the LED spectrum was previously recorded by directly irradiating the LED light to the spectrometers. After proper normalization, accounting for the reflection coefficients of the fiber ends, forming the interferometer, the quasi-static component of the both spectra was removed. This allowed to accurately estimate the positions of the FPI spectrum fringes by fitting the spectrum peaks by Gaussian functions. The spectra obtained with the commercial spectrometer and SPBS are shown in Figure 3a,b, respectively. Different envelopes of the spectra are due to the additional filtering performed by the red, green and blue channels of the smartphone camera; however, since we were interested only in fringe positions, the envelopes did not affect the calibration results.

The correspondence between the peaks positions is shown in Figure 4 together with the linear fit. The positions of the red-to-green and green-to-blue transition areas of SPBS spectrum were estimated as described in the previous subsection, their corresponding wavelengths were found using the linear fit of FPI peaks’ coordinates.

Wavelength sampling interval of the SPBS (the difference between the coordinates of two adjacent points of the measured spectrum, not to be confused with the spectral resolution, which is a width of an instrument function) could be evaluated after this calibration and turned out to be 0.22 nm/pixel. Spectral resolution of the assembled smartphone spectrometer was also experimentally evaluated by observing the spectra of 3 lasers: green and red semiconductor lasers with spectral linewidths $\Delta \lambda_{\mathrm{SOURCE}}$ around 4 nm and Thorlabs HNL210LB He-Ne laser with overall spectral width around 1.9 pm (1.5 GHz in terms of frequency). Due to the above-mentioned convolution of the initial spectrum with the instrument function, the width of the measured spectrum $\Delta \lambda_{\mathrm{MEAS}}$ can be expressed as [36]
(6)$$\Delta \lambda_{\mathrm{MEAS}}=\sqrt{\delta \lambda^{2}+\Delta \lambda_{\mathrm{SOURCE}}^{2}},$$
where δλ is the spectral resolution of the spectrometer (typically introduced as full width at half maximum (FWHM) of its instrument function). The widths of the measured spectra $\Delta \lambda_{\mathrm{MEAS}}$ were 5.8 nm, 5.7 nm and 4.1 nm for green semiconductor, red semiconductor and He-Ne lasers, respectively. Therefore, spectral resolution of the assembled spectrometer turned out to be 4 nm.

This calibration operation needs to be performed only once prior to the use of the SPBS for optical fiber sensor interrogation if the spectrometer IF and the spectral transfer functions of the RGB channels of the smartphone camera do not change over time.

### 3.4. Interferometer Optical Path Difference Demodulation

Due to nonuniformity of the LED spectrum, the measured interferometer’s spectra have a quasi-constant additive component and parasitic amplitude modulation. The most convenient way to eliminate these signal distortions is their cancellation based on the a-priori information on the LED spectral shape. This way, preceded by proper normalization, the LED spectrum can be subtracted from the measured interferometer spectrum, thus canceling the quasi-static component. Parasitic amplitude modulation can be suppressed by normalizing the interferometer spectra on the LED spectrum.

After the quasi-static component and parasitic amplitude modulation were suppressed, the interferometer spectra were processed, resulting in demodulation of the interferometer OPD $L_{0}$. A least-squares approach, described in detail in [39] was used to estimate the OPD. This approach uses fitting of the measured spectrum with an analytical equation of a form
(7)S(L,λ,φ)=cos(4πL/λ+φ),
where *L* is interferometer OPD and is the optimization parameter, λ is the wavelength scale, calculated using the hue-based self-calibration approach described above, φ is an additional phase that is often non-zero in extrinsic and in-fiber interferometers due to light diffraction during free space propagation [40] and mode coupling effects [41]. It must be noted that ignorance of this phase shift or its incorrect accounting may result in abrupt λ/2 errors [42]. In our case of extrinsic fiber Fabry-Perot interferometric sensor, additional phase shift was primarily caused by the diffractive broadening of the optical beam inside the cavity, a phenomenon also known as Gouy phase. Strict analysis of diffractive broadening of a beam exiting multimode fiber is quite complicated and falls out of the scope of the current paper. Therefore, the form $\varphi =\mathrm{atan}(L\lambda /\left(\pi w_{0}^{2}\right))$, where $w_{0}$ is the fiber core diameter was used as an approximation, which is valid for a singlemode fiber output. Anyway, thanks to the small wavelength and short cavity length of the used EFPI, this phase shift is quite small and even its not exact accounting will not cause any critical errors.

The whole signal processing workflow, whose stages were described above, including spectrum image cropping, hue transition areas identification, spectra interpolation, averaging and normalization, as well as the wavelength scale calculation and interferometer OPD demodulation is shown schematically in Figure 5. The signal processing was designed and at first tested in Matlab on a personal computer. After the initial tests, the developed Matlab programs were ported to the smartphone, which has the Matlab Mobile application installed.

## 4. Experimental Results

The performance of the developed smartphone-based OFS interrogation system and the proposed spectral self-calibration approach was verified experimentally, for which we have carried out two experiments. The first one was aimed at proving the ability of plug-and-play operation, i.e., provide accurate interferometer OPD readings after reattachment of the fiber and diffraction grating holder from the smartphone without any calibration. The second experiment was aimed at demonstrating an improvement of the temperature stability of the smartphone-based OFS interrogation system.

### 4.1. Verification of Plug-And-Play Operation

Obviously, the wavelength-to-pixel number calibration of a smartphone-based spectrometer with conventional pre-calibration is lost after the fiber and diffraction grating holders are shifted or removed and reattached. Given that the size of the state-of-the-art smartphone camera pixels is on the order of 1–2 μm, micrometer-level movements of the fiber and diffraction grating holders will result in significant change of the wavelength scale. Therefore, plug-and-play operation of the smartphone-based spectrometers requires either to permanently incorporate the smartphone into the common holder, or can not be performed without auxiliary external light sources with well-defined spectra.

Our hue-based spectral self-calibration approach allows us to overcome this limitation and enables correct wavelength scale reconstruction and accurate signal demodulation in case of reasonably large offsets of the fiber and diffraction grating holders. A series of experimental trials was performed to verify this ability. The interferometric sensing element was stabilized by being put inside a thermally and vibrationally isolated chamber, thus ensuring that its true OPD remained constant, while fluctuation of the measured OPD value can characterize the robustness and accuracy of the proposed spectral self-calibration approach. Prior to each measurement of the interferometer spectrum, the holder was removed from the smartphone and then reattached without any special alignment. The only condition for the calibration and the measurement procedures to be performed is that the light from the LED is coupled to the fiber and interferometer spectrum is visible at some place of the camera image. Six spectra images are shown in Figure 6a–f. Horizontal, vertical and small angular misalignment of spectra can be clearly noticed.

However, despite noticeable misalignment of the measured spectra images, the developed spectral self-calibration approach enables to reduce the detrimental effect of misalignment and provide an accurate demodulated sensor signal. This is shown in Figure 7, demonstrating demodulated OPD values of stabilized interferometer, corresponding to the spectra images shown in Figure 6. It can be seen that the wavelength scale of the spectra is reconstructed with high accuracy—the standard deviation of the demodulated OPD values was about 0.6 nm, while the deviation of the wavelength scale shift was less than 0.2 nm. On the other hand, as can be seen in Figure 6, the initial vertical shifts of spectra images are around 100 pixels, which corresponds to more than 20 nm. Therefore, more than two orders of magnitude reduction of the wavelength scale shift was achieved by applying the proposed self-calibration approach.

### 4.2. Verification of Temperature Stability

It is logical to anticipate that thermal expansion of the 3D-printed holders, which provide the alignment of all optical elements (LED, fibers, diffraction grating and the camera lens) will cause the change of the initial elements alignment. In turn, this will lead to the shift of the optical spectrum projected onto the smartphone camera and change of the corresponding wavelength scale. Therefore, the initial spectral calibration (correspondence between the pixel number and optical wavelength) will be lost, resulting in incorrect demodulation of sensors interrogated in spectral domain, such as interferometric and fiber grating-based sensors.

Our proposed spectral self-calibration approach is aimed at solving this problem. The verification experiment was carried out in the following manner—the smartphone-based OFS interrogation system was turned on and the temperature on the smartphone surface near the fiber and diffraction grating holder was monitored with an external electronic thermometer with a small footprint. The smartphone itself produces some heat even in idle mode; moreover, when the interrogation system is operating and there is an increased workload on the processor and the LED is switched on, it results in heating of the holder, causing it to expand, which results in a shift of the measured spectra images.

On the other hand, if the interrogated interferometer is placed in a thermally and mechanically isolated chamber, its real OPD remains constant, while the change of the demodulated OPD value will clearly indicate the influence of the interrogation system heating on the measurement outcome.

During the conducted experiment, the flash LED was constantly turned on, resulting in gradual change of temperature on the smartphone back panel near the 3D-printed fiber holders from nearly 33 Celsius to almost 41 Celsius over about an hour, which is shown in Figure 8a. The curve shows exponential fit of the experimental points, corresponding to Newton’s heating law. Minor discrepancies between the points and the fit are likely to be caused by the change of the processor workload, network activity of background processes and other reasons. Anyway, the plot in Figure 8a clearly shows that even in case of constant ambient conditions, the temperature of SPBS might vary significantly.

Demodulated OPD values obtained with and without the developed hue-based spectral self-calibration approach as well as the shifts of the wavelength scale are shown in Figure 8b. Zero wavelength shift corresponds to initial spectrum position. It can be clearly seen that, without the spectral self-calibration, the shift of the wavelength scale follows the temperature deviation of the interrogation system, demonstrating about 37.5 pm/C dependency. On the other hand, when the wavelength scale recalculation was introduced, the temperature drift was eliminated, at least on the level of demodulated OPD fluctuations. Relatively long realization of measured spectra allowed to estimate the standard deviation of demodulated OPD values, which turned out to be around 1 nm (with corresponding standard deviation of the wavelength scale shift around 60 pm) and is shown with dashed lines in Figure 8b as well as the mean OPD value.

## 5. Conclusions

The paper presents a novel spectral self-calibration approach for smartphone spectrometers. Thanks to the mutual overlap between the spectral ranges covered by the red and green and green and blue channels of a smartphone camera, the hue distribution of the measured spectrum has two distinctive features, whose corresponding wavelengths can be calibrated with a reference spectrometer. Once the wavelengths corresponding to the hue distribution features are known, these features can be further used to restore the wavelength scale of the measured spectrum.

The use of the developed spectral self-calibration approach allowed to eliminate temperature dependence of the smartphone spectrometer readings and make the system plug-and-play, in contrast to the most other known smartphone spectrometers, which spectral accuracy relies upon wavelength-to-pixel calibration performed with external reference light sources and is lost if the fiber and diffraction grating holders are removed and reattached to the smartphone, for example, during transportation. However, the fallout of this improved accuracy is about 10 times reduced OPD measurement resolution, which is a result of finite accuracy of fitting the slopes of hue distribution. However, in practical situations self-calibration can be performed not continuously, but upon user’s request or periodically in order to detect deviations of the wavelength scale. This will enable to achieve a trade-off between the sensing resolution and accuracy.

The performed experiments validated the efficiency of the developed spectral self-calibration approach and its ability to be used for interrogation of optical fiber Fabry-Perot interferometers. Further work must be directed towards estimating the accuracy limitations of the proposed approach (induced by the spectrum noises, finite spectral resolution of the smartphone spectrometer, shapes of spectral transfer functions of optical filters used in red, green and blue channels of the smartphone camera and others) and ways to improve the accuracy. It should be noted that the developed spectral self-calibration approach can be applied not only to interferometric sensors, but also to other sensors with spectral interrogation, such SPR and some grating-based sensors, as well as to spectroscopic tasks. The main requirement for robust operation of the proposed spectral self-calibration approach is relatively high spectrum intensity in red-to-green and green-to-blue transition areas, finite mutual overlap of red and green and green and blue camera channels’ spectral functions, resulting in nonuniform hue distribution of the measured spectrum image and relatively high spectrometer spectral resolution ensuring correct capture of the spectra images’ hue distribution.

Obtained experimental data, Matlab source codes for spectra images’ processing and models for 3D-printing the fiber and diffraction grating holder are available from the authors on reasonable request.

## Figures and Tables

**Figure 1 sensors-20-06304-f001:**
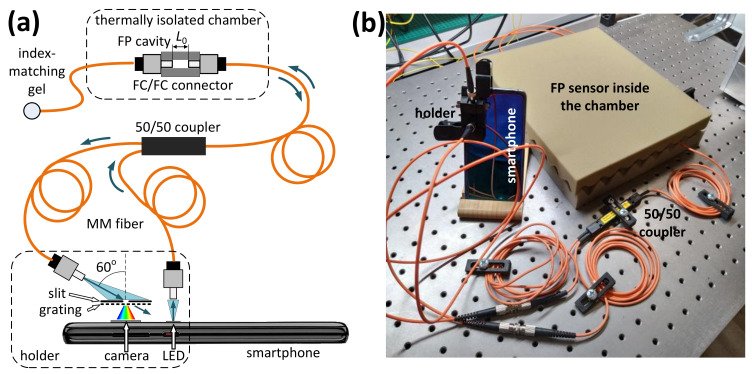
The schematic overview (**a**) and the photograph (**b**) of the developed smartphone-based spectrometer and its application for optical fiber Fabry-Perot interferometric sensor interrogation.

**Figure 2 sensors-20-06304-f002:**
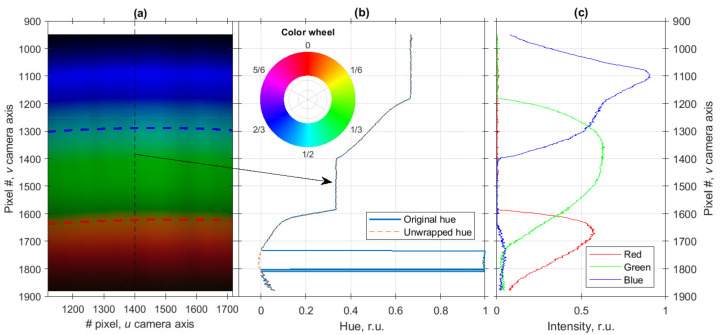
Raw image of the smartphone LED spectrum measured with the smartphone-based spectrometer, the centers of the transition areas are shown with red and blue dashed curves (**a**); corresponding hue distribution along the *v* axis (**b**); red, green and blue channels’ intensities (**c**). The color wheel in the inset of (**b**) is shown to demonstrate the transformation between the RGB format and hue, as well as the analogy between the hue and the phase (or angle).

**Figure 3 sensors-20-06304-f003:**
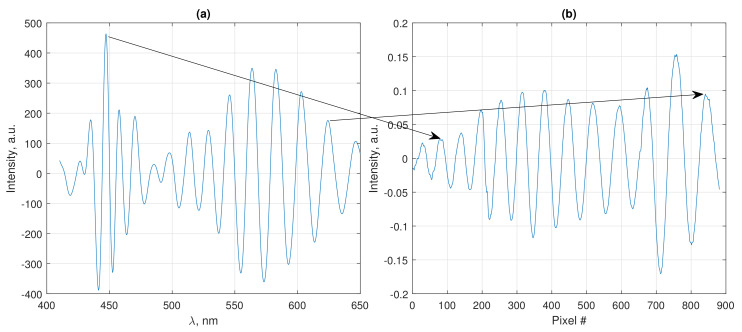
Comparison of stable Fabry-Perot interferometer (FPI) spectra measured with commercial spectrometer (**a**) and smartphone-based spectrometer (SPBS) (**b**). Arrows show the first and the last pairs of corresponding peaks, used for calibration.

**Figure 4 sensors-20-06304-f004:**
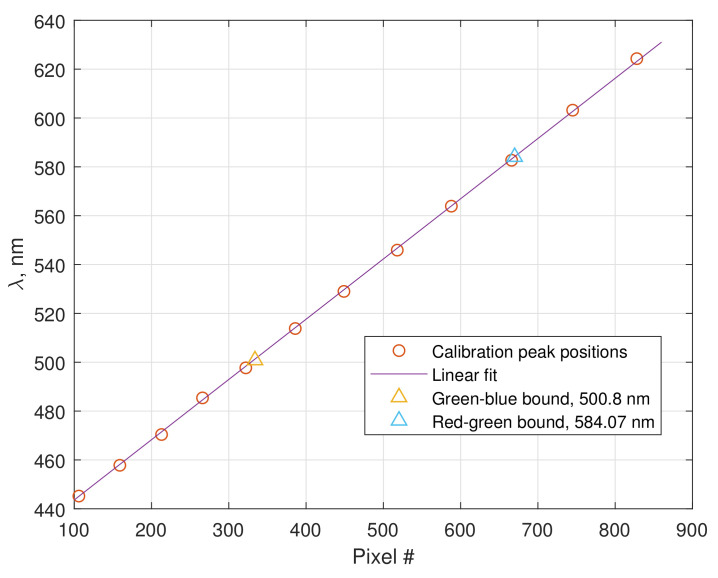
Wavelength calibration curve.

**Figure 5 sensors-20-06304-f005:**
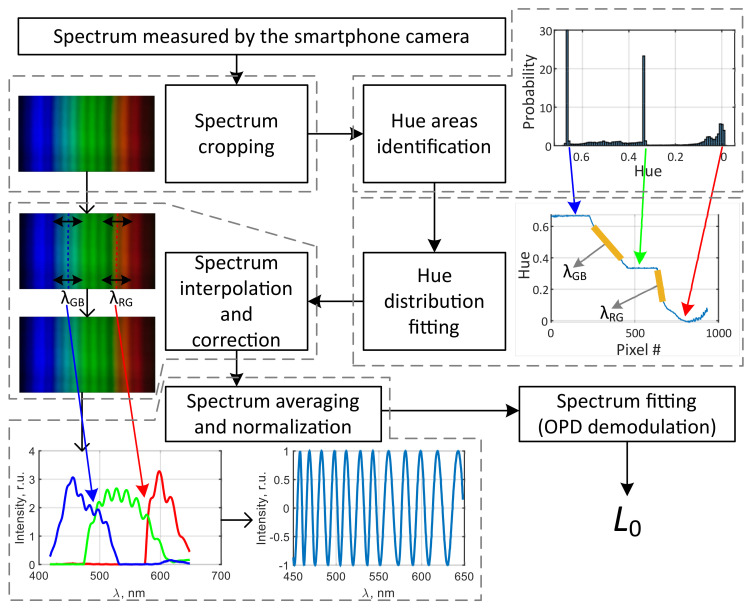
Schematic representation of signal processing workflow. Dashed gray lines indicate separate processing stages and what changes are made to the signals.

**Figure 6 sensors-20-06304-f006:**
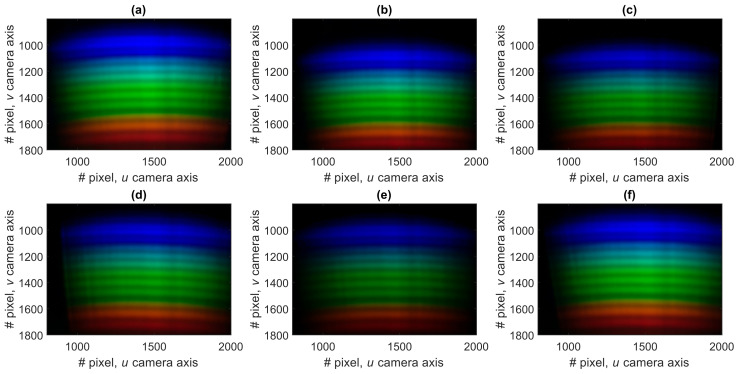
Spectra measured when the fiber and diffraction grating holders were removed from the smartphone and reattached right between the measurements. In (**a**–**f**) different spectra images, obtained during consecutive measurements, are shown.

**Figure 7 sensors-20-06304-f007:**
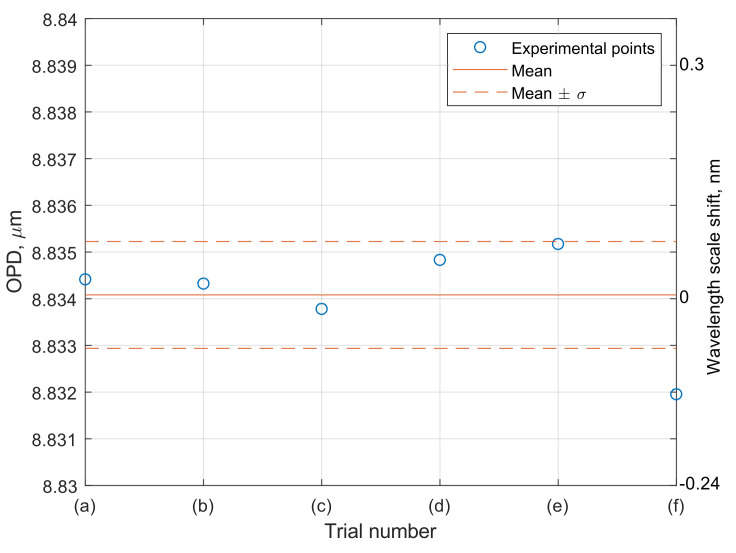
Interferometer optical path differences (OPD) values (left vertical axis) and corresponding spectra wavelength shifts (right vertical axis) demodulated from the spectra shown in Figure 6a–f. Solid line shows the mean OPD and dashed lines show the standard deviations of the experimental data.

**Figure 8 sensors-20-06304-f008:**
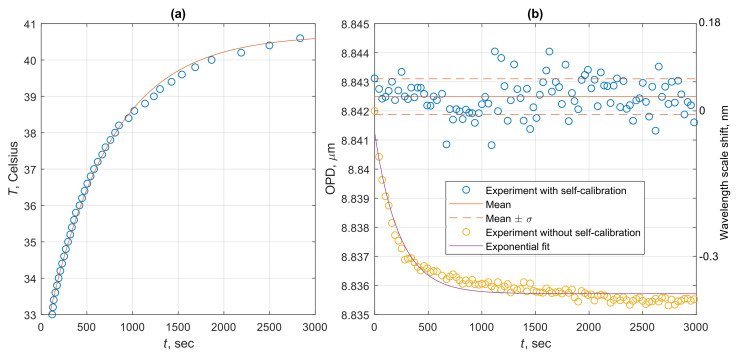
(**a**) Change of the temperature on the smartphone surface during the optical fiber sensor interrogation. Circles—experimental points, curve—exponential fit. (**b**) Curves of demodulated interferometer OPD in case of application of the proposed spectral self-calibration and conventional demodulation with only initial wavelength calibration. Right-hand side vertical axis illustrates the corresponding shift of the wavelength scale. Solid line shows the mean OPD and dashed lines show the standard deviation of the experimental data.

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
