# Peer review of "Continuous Hue-Based Self-Calibration of a Smartphone Spectrometer Applied to Optical Fiber Fabry-Perot Sensor Interrogation"

_sensors, 2020, doi:10.3390/s20216304_

Round 1
Reviewer 1 Report
The paper is surely interesting and timely. The background is clearly presented and discussed. The structure of the paper is well organized, and the methodology is robust and comprehensive. English grammar is fine. I have not found any scientific flaws. The only marginal comment is about the applicability to this SPBS for FBG sensing, although the Authors published a paper about that: the resolution of 4 nm is way too coarse to most practical applications.
In my opinion, the paper can be accepted as it is.
As a final comment, it would be nice to release the code of the app and the STL files for the holder.
Author Response
The paper is surely interesting and timely. The background is clearly presented and discussed. The structure of the paper is well organized, and the methodology is robust and comprehensive. English grammar is fine. I have not found any scientific flaws. The only marginal comment is about the applicability to this SPBS for FBG sensing, although the Authors published a paper about that: the resolution of 4 nm is way too coarse to most practical applications.
In my opinion, the paper can be accepted as it is.
As a final comment, it would be nice to release the code of the app and the STL files for the holder.
Reply:
We thank the Reviewer for evaluation of our work. As regards releasing the processing code and STL model, we wouldn't like to make them freely available, although we will be happy to share them with those who clearly declare and justify their interest. We have added a sentence in the Conclusions that the experimental data, source code and STL models are available from the authors on reasonable request.
Reviewer 2 Report
The manuscript is very comprehensive, with an original solution for the calibration of smartphone spectrometers. The practical benefit from this approach compared to conventional method using an external light source is still in doubt, because of the drop in the measurement resolution, but nevertheless is a good original contribution.
The title of the article is somewhat misleading. because the main contents of the article is the self-calibration. A small part of the paper is devoted to the use of this system for experiments involved interrogation of a optical fiber Fabry-Perot interferometric sensor. The abstract is correctly written.
English grammar need to be revised
Author Response
The manuscript is very comprehensive, with an original solution for the calibration of smartphone spectrometers. The practical benefit from this approach compared to conventional method using an external light source is still in doubt, because of the drop in the measurement resolution, but nevertheless is a good original contribution.
Reply:
We thank the Reviewer for evaluation of our work. Indeed, the sensing resolution when the proposed self-calibration method is decreased. However, in practical situations, it can be used not all the time, but only after attachment of fiber and diffraction grating holder or periodically to monitor if there is any wavelength scale drift due to the smartphone heating/cooling. In this scenario, the resolution will remain the same, while the sensor accuracy will be improved compared to the case without self-calibration. This was pointed out in the Conclusions of the revised manuscript. This and other modifications are marked with red font.
The title of the article is somewhat misleading. because the main contents of the article is the self-calibration. A small part of the paper is devoted to the use of this system for experiments involved interrogation of a optical fiber Fabry-Perot interferometric sensor. The abstract is correctly written.
Reply:
We are afraid we tend to disagree with the Reviewer. Indeed, the proposed self-calibration approach is the central subject under consideration, therefore the title is starting "Continuous Hue-Based Self-Calibration ...". However, the accuracy of interferometric optical fiber sensors is especially affected by the wavelength scale miscalibration, therefore, such an approach is of particular practical importance for this type of sensors. Moreover, as mentioned at the end of Section 3.1 and in the Conclusions, there are several requirements that must be fulfilled in order to make the proposed self-calibration approach applicable. Interferometric sensors are the ones for which these conditions are fulfilled. Moreover, we didn't want the title to be too broad and general, so that the readers are not confused with its possible applications.
English grammar need to be revised
Reply:
While writing the manuscript, we consulted with a colleague who has a proficient level of English and is experienced in scientific writing. However, while proofreading the manuscript during revision, we have found and corrected some misprints. Also some phrases were reformulated. We thank the Reviewer for pointing this out and hope that the readability of the revised manuscript was improved.